

# T cell receptor repertoire as a novel indicator for identification and immune surveillance of patients with severe obstructive sleep apnea

Kai Li[1,2,3], Yue Zhuo[4], Yue He[5], Fei Lei[6], Pengming He[4], Qin Lang[7], Dingxiu He[1,8], Suni Zuo[3], Shan Chen[2], Xin Yang[4], Xueping Wen[9], Zhixin Zhang[4,9] and Chuntao Liu[1]

[1] Department of Respiratory and Critical Care Medicine, West China Hospital, Sichuan University, Chengdu, China
[2] Department of Respiratory and Critical Care Medicine, The Third People's Hospital of Chengdu, Southwest Jiaotong University, Chengdu, China
[3] Department of Respiratory Medicine, The People's Hospital of Pujiang County, Chengdu, China
[4] Department of Health Management & Institute of Health Management, Sichuan Provincial People's Hospital, University of Electronic Science and Technology of China, Chengdu, China
[5] Department of Orthopedics, West China Hospital, Sichuan University, Chengdu, China
[6] Sleep Medicine Centre, West China Hospital, Sichuan University, Chengdu, China
[7] Department of Pulmonary Diseases and Critical Care, Sichuan Provincial People's Hospital, University of Electronic Science and Technology of China, Chengdu, China
[8] Department of Emergency Medicine, The People's Hospital of Deyang, Deyang, China
[9] Chengdu ExAb Biotechnology LTD, Chengdu, China

Corresponding author
Chuntao Liu, taosen666999@163.com

## ABSTRACT

**Background.** Obstructive sleep apnea (OSA) is the most prevalent sleep disturbance that affects approximately 936 million people worldwide and leads to extensively increased incidence of cardiovascular disease, metabolic syndrome, neurological disorders, and traffic accidents. Severe OSA patients suffer a significantly higher risk of complications and worse comorbidity outcomes. Notwithstanding, with inadequate access to contact diagnosis based on polysomnography (PSG), numerous patients with severe sleep apnea have not been diagnosed, especially during the pandemic. Moreover, how the T cell immunity is impaired in OSA remains largely unknown.

**Methods.** We primarily investigated the T cell receptor (TCR) repertoires of 50 patients with severe OSA, 23 patients with mild-to-moderate OSA, 23 patients without OSA, and 157 healthy individuals, from their peripheral blood. Firstly, we compared the clinical characteristics, blood cell counts, the ratio of neutrophil-to-lymphocyte (NLR), platelet-to-lymphocyte (PLR), and CD4+/CD8+T cell count between groups. Then, we compared the diversity, clonotypes, unique VJ alleles in patients with different disease severity. Furthermore, by identifying a series of disease-associated amino acid sequences, we employed a repeated hold-out machine learning strategy to explore the optimal algorithm for calculating the TCR repertoire characteristic Index (OSA-TCI). We further confirmed its relation with clinical features by linear regression analysis. Moreover, in followup of severe OSA patients who accepted adherent non-invasive ventilation, we assessed the changes of TCR repertoires, OSA-TCI, ESS, NLR, PLR, and CD4+/CD8+T after therapy.

**Results**. We found an unexpected increase in diversity and clonotypes in the TCR repertoire of OSA patients. Furthermore, we successfully developed a novel indicator termed OSA-TCI to summarize the unique repertoire alteration, which provided 90% of sensitivity and 87% of specificity in distinguishing severe OSA. In rationalization, OSA-TCI was found correlated to AHI, BMI, hemoglobin, N1, N2 percentage of sleep, snoring, smoking and lowest oxygen saturation, but only independently related to AHI ($R = 0.603$) and smoking ($R = 0.22$). Finally, we observed OSA-TCI in the eight severe patients decreased significantly after home noninvasive ventilation for three months during follow-up, consistently in line with the TCR repertoire improvement. In contrast, NLR, PLR, and the ratio of CD4+/CD8+T cell count were found useless to diagnose and therapeutic surveillance of severe OSA.

**Conclusions**. Our study is the first to unveil the TCR repertoire alteration in OSA, indicates possible insidious autoimmune mechanisms underlying OSA, and suggests that TCR repertoires serve as a convenient peripheral blood biomarker for OSA assessment without long-time contact and facility/instrument occupation. It may shed light on future diagnostic, immunological, pathophysiological, and prognostic research on OSA.

# INTRODUCTION

Obstructive sleep apnea (OSA), affecting up to 936 million people worldwide, of which 425 million are moderate to severe patients, is the most prevalent sleep disturbance with a high social and economic burden (*Benjafield et al., 2019*). Untreated patients with OSA are at higher risk (*George, 2007*; *McNicholas et al., 2018*; *Bradley & Floras, 2009*; *Justeau et al., 2020*) of metabolic syndrome and type 2 diabetes, cardiovascular and cerebrovascular morbidity, neuropsychiatric dysfunction, pulmonary hypertension, cancer, motor vehicle accidents, and even worse co-morbidity prognosis, including severe COVID-19 (*Rögnvaldsson et al., 2022*). Chronic inflammation is one of its most essential pathological features (*Lavie, 2015*), but the role of adaptive immunity and how it is altered in OSA remains unclear. Unfortunately, with inadequate access to diagnosis based on polysomnography (PSG), numerous patients with moderate to severe OSA have still not been diagnosed (*McNicholas et al., 2018*), especially during the pandemic.

In OSA, chronic low-grade inflammation and immune impairment are widely documented and are suspected to be responsible for systemic injury, complications, and worse prognosis (*Bradley & Floras, 2009*; *Lavie, 2015*). Studies revealed a series of changes from cytokines, chemokines, and metabolic products to blood cell categories in OSA (*Yokoe et al., 2003*; *Ryan, Taylor & McNicholas, 2005*; *Cubillos-Zapata et al., 2017*), as an increase in CRP, IL-6, IL-10, TNF-a, NFkB, PD-1, cell-free DNA, and neutrophil/lymphocyte ratio (NLR) (*Rha et al., 2020*), CD4+T/CD8+T ratio (*Domagała-Kulawik et al., 2015*). The

macrophages population was also shifted towards the M1 subset (*Khalyfa, Kheirandish-Gozal & Gozal, 2018*). There is proceeding evidence ahead of the immune mechanism nature of OSA, but little is known about the initiation and linkage of the cellular immune response till now, particularly, the T cell activation and proliferation.

As is known to all, T cells are at the core of the adaptive immune in humans, responsible for infection immunity, autoimmunity, tumor immunity, and immune regulation. While its activation and proliferation depend on the combination of T cell receptor (TCR) to a corresponding MHC-antigen peptide complex (pMHC) on antigen presenting cells (APCs) (*Rajewsky, 1993*). Theoretically, there could be over $1 \times 10^{13}$ possible TCR types to recognize and combat almost unlimited types of possible antigens (*Nielsen & Boyd, 2018*). Thus, the tremendously diversified repertoires of TCR is a snapshot of an individual's T cell immune status, which embodies information about exposure to endogenous and exogenous antigen presented by APCs, the ability of adaptive immune system to resist new challenges, as well as the character, extent, memory of immune responses. The most variable region of TCR, the complementary determinant region 3 (CDR3), which is critical in determining antigen specificity, is generated through variant rearrangements of different V, D, and J gene segments, in addition to the template nucleotide insertion and deletion, can be a representative for each T-cell clone (*Woodsworth, Castellarin & Holt, 2013*). High throughput sequencing of TCR beta CDR3 alone allows to reflect directly the T-cell directory in considerable measure.

Our study, with the aim to investigate TCR repertoires alterations in peripheral blood of OSA patients and to explore potential biomarkers for severe OSA identification and assessment, found a significant increase in diversity and clonotypes in OSA patients. Further analysis of V, J gene usage and amino acid clonotypes bias confirmed the correlation with OSA severity. Then we developed a novel indicator termed OSA TCR Characteristic Index (OSA-TCI) to summarize the extent of disease-associated TCR repertoire changes, which was subsequently validated as accurate in distinguishing patients with severe OSA from others. Moreover, we observed a consistent decrease in OSA-TCI and improvement of TCR repertoire in eight severe patients after treatment during follow-up. These findings may improve our understanding of OSA's cellular immune function impairment and provide us with a convenient peripheral blood biomarker for severe OSA assessment without long-time contact or facility/instrument occupation. In addition, it may shed light on future diagnostic, immunological, pathophysiological, and prognostic research on OSA.

## MATERIAL AND METHODS

### Subjects and ethic approval

Adult participants who newly finished overnight PSG within one month in the Sleep Medical Centre of West China Hospital, Sichuan University, were initially recruited between September 2021 and April 2022, with symptoms suggesting potential sleep-related breathing disorders. TCR CDR3 repertoire data of 157 concurrent sex-and-year matched healthy donors from the database of Health Management Department, Sichuan Provincial People's Hospital were enrolled as healthy control.

**Exclusive criteria:** Patients with a history of chronic inflammatory or allergic disease, including active immune deficiency, autoimmune disease, renal failure, tumor, thyroid dysfunction, recent infection, trauma, vaccination, bone marrow transplantation, or any invasive medical procedure within three months were eliminated. Patients with respiratory failure, heart failure, uncontrolled hypertension, hematonosis, morbid obesity (body mass index (BMI) >35 kg/m²), stroke, anxiety or depression on medication, taking reverse transcription inhibitors or immunomodulators were ineligible for our study, too. Heavy smokers (smoking >20 cigarettes per day), night shift workers, and patients who suffer from or coexist with other sleep disorders including insomnia, narcolepsy, central sleep apnea syndrome, or periodic limb movement were also removed from the candidates according to PSG results.

The experimental design and participant recruitments were approved by the Ethics Committee of West China Hospital, Sichuan University (Ethics Approval for Research: 2021 #1302) and the Medical Ethics Committee of Sichuan Provincial People's Hospital (Ethics Approval for Research: 2020 #351). Prior to enrollment, all patients confirmed their informed consent in written form. The flowchart of our study algorithm can refer to Fig. S1.

## Sleep study by PSG

Each subject finished overnight PSG >7 h with the Philips Alice 6 Sleep System (Philips, Amsterdam, The Netherlands), which recorded EEG, electro-oculography, electro-myography, ECG, nasal airflow pressure and temperature, pulse oximetry, microphone for snoring monitoring, thoracic and abdominal movements, body position and leg movements. A single registered sleep technologist completed all manual scores of PSG recordings for each subject. The definition of apnea, hypopnea, RERAs, and apnea/hypopnea index (AHI) were strictly followed by the guideline for sleep disorders, AASM 2017 (*Kapur et al., 2017*). The AHI was defined as the hourly frequency of apnea and hypopnea events in total sleep time (TST). The severity of hypoxia was evaluated by the lowest oxygen saturation and total duration with oxygen saturation lower than 90% during sleep. A senior sleep doctor made the diagnosis and obeyed the diagnostic criteria and severity classification of sleep disorders. Accordingly, participants were classified as non-OSA (AHI <5), mild-to-moderate OSA (5 <AHI <30) and severe OSA (AHI ≥ 30).

## Preparation of TCRβ repertoire from peripheral blood samples

Peripheral blood (PB) samples were obtained between 7:30 and 9:00 in the morning after a one-night fast in West China Hospital. Along with PB samples for blood cell routine analysis, T cell sub-types count, antibodies, and whole blood samples were collected for PBMCs preparation. The whole procedure of PBMCs preparation, RNA purification and amplification, and TCR repertoire sequencing were completed in the laboratory of Health Management Institute, Sichuan Provincial People's Hospital, while other PB samples were examined at the West China Hospital.

Following the protocol we previously published (*Zhuo et al., 2022*), TCRβ variable region genes were amplified through RT-PCR using RNA extracted from peripheral blood

mononuclear cells (PBMCs) as a template without previous T cell subsets isolation, and then sequenced.

Sequencing results were processed and a TCRβ repertoire containing 30,000 functional TCRβ sequences, whose V-D-J junction produced a productive translation of TCRβ peptide with a non-empty CDR3 region, was prepared for each sample. The diversity of a TCRβ repertoire was measured by the diversity 50 (D50) value and the Shannon's diversity index.

## Model development and validation

**Step 1**: A total of 20 out of all 157 healthy donor samples were randomly allocated to the testing control group, while the remaining 137 samples were arranged to the modeling control group. Correspondingly, 9 out of all 50 severe OSA patient samples were randomly allocated to the positive testing group, while the other 41 samples were left to the positive modeling group.

**Step 2**: A total of 103 samples were randomly picked out of the modeling control group (75% of 137) and allocated to the training control group, and the remaining 34 samples were enrolled in the validating control group. Meanwhile, 33 samples were randomly picked out of the positive modeling group (80% of 41) and allocated to the positive training group, while the other eight were enrolled in the validating positive group. All unique TCR CDR3 sequences that were ever presented in the repertoires of samples in the modeling control group or the positive modeling group were included in the modeling TCR set.

**Step 3**: All TCR CDR3 sequences with >4% commonality (presented in repertoires of >4% samples in the group) in the training control group were removed from the modeling TCR set. Then, all TCR CDR3 sequences with <2% commonality in the positive training group were removed from the modeling TCR set. Additionally, all TCR CDR3 sequences whose average frequency in the training control group is 10 times greater than in the positive training group were removed from the modeling TCR set. Subsequently, the remaining TCR CDR3 sequences in the modeling TCR set are defined as the OSA-TCR sequences. The OSA-TCR Characteristic Index (OSA-TCI) for each sample was defined as the sum of frequencies for all OSA- TCR sequences in the repertoire.

**Step 4**: The receiver operating curve (ROC) was drawn according to the OSA-TCI of each sample in the validating control group and validating positive group. We calculated the area under the ROC curve (AUC).

**Step 5**: Repeat steps 2 to step 4 for 20 times. When the modeling achieved the highest AUC value, it was recorded as the optimal model for OSA-TCI.

Finally, we calculated the TCI value for each sample in the previously left-alone testing groups by applying the optimal model and then drew the prognostic ROC as external validation. The flowchart of our procedures is available in Fig. S2.

## Rationalize OSA-TCI with clinical features and follow-up

In all, 96 samples with complete PSG data, we performed a linear regression analysis to explore independent factors associated with OSA-TCI that may help us rationalize the relationship between OSA-TCI and clinical characteristics. To assess the association with OSA-TCI, we also calculated the correlation coefficient for each independent factor. A

partial correlation coefficient analysis was subsequently employed to clarify the independent association aside from confounding factors.

A consecutive follow-up cohort covering all 50 severe OSA in the study has been initiated since the enrollment in our study. Either treatment or complications may impact the TCR repertoire of OSA. Therefore, in the follow-up, much of our attention is paid to their longitudinal TCR repertoire, clinical symptoms, the score of Epworth Sleepiness Scale (ESS), comorbidities, and complications 3 months after the nocturnal treatment with positive airway pressure (nPAP) to verify further the value of the TCR repertoire in therapeutic efficacy monitoring of severe OSA.

## Statistical analysis

All statistical analysis was performed by the ORIGIN Pro software version 2021 (OriginLab Corporation). The measurement data were first tested for normality. Normal distributive data of continuous variables were described as mean $\pm$ standard deviation format. One-way ANOVA was employed in multiple group comparison: when variance was combined with the overall comparison using the F-test, the Bonferroni method was used for multiple comparisons; For unequal variances, the Welch's approximate F-test was used, and the Dunnett's T3 method was utilized for comparisons in multiple groups. Non-normal distributed Indexes were described by median (interquartile interval). Non-parametric Kruskal-Wallis test was utilized to compare multiple sets of independent variables. In addition, the Wilcoxon matched-pairs signed ranks test was employed to compare two sets of dependent variables. Between groups, the Chi-square test was applied to compare the counting indexes. Finally, linear regression and Spearman correlation analysis were employed to explore the correlation between OSA-TCI and clinical characteristics. The difference of $P < 0.05$ was considered statistically significant.

# RESULTS

## Demographic and clinical characteristics

Totally 50 patients were diagnosed with severe OSA, 23 were diagnosed with mild-to-moderate OSA (eight mild and 15 moderate), and 23 age-and-sex matched subjects with AHI < 5 excluded from the diagnosis of OSA were classified to the non-OSA group in this study. Their demographic and clinical characteristics, blood cell counts, level of antibodies, and primary PSG metrics are summarized in Table 1. AHI, BMI, ESS score, smoking rate, blood pressure, micro-arousal Index, and hemoglobin concentration were significantly higher in patients with severe OSA. No differences were found statistically significant between any two groups from the standpoint of age, gender, total sleep time, sleep structure, monocyte, platelets, antibodies, T lymphocyte categories, and the CD4+/CD8+T ratio. While the counts of white blood cells and neutrophils in OSA patients showed an increasing trend with statistical differences, the difference in NLR and platelet/lymphocyte ratio (PLR) between groups was insignificant.

## Repertoire diversity and clonality in severe OSA

We analyzed the diversity and abundance of TCR clonotypes. The D50 diversity index and Shannon's diversity index in both OSA groups were unexpectedly significantly higher

**Table 1  Demographic and clinical characteristics of the study groups.**

| | Healthy donors | Non-OSA | Mild-to-Moderate OSA | Severe OSA | _P_ value |
|---|---|---|---|---|---|
| **Subjects n** | 157 | 23 | 23 | 50 | |
| **Gender** | 95/62 | 15/8 | 15/8 | 36/14 | 0.527 |
| **Age Years** | 39(32, 50) | 32(29, 44) | 37(32, 49) | 40(33, 52) | 0.226 |
| **Body mass index kg m$^{-2}$** | 22.02(20.26, 23.18)[c,d] | 22.55(21.05, 24.65)[d] | 24.93(21.88, 25.83)[a] | 27.04(24.33, 29.74)[a,b] | **0.000** |
| **Current Smokers n (%)** | 13(8.28)[d] | 4(16.67)[d] | 1(4.76)[d] | 16(34.0)[a,b,c] | **0.000** |
| **Epworth sleepiness scale** | / | 4(2, 7) | 6(2, 8) | 6(3, 10.25) | 0.485 |
| AHI events h $-1$ | / | 3.20(1.80, 4.10)[c,d] | 17.80(10.50, 22.70)[b,d] | 58.95(44.80, 75.95)[b,c] | **0.000** |
| Oxygen desaturation index, events h $-1$ | / | 1.4(0.9, 2.5)[c] | 13.0(3.8, 18.9)[c] | 56.3(38.0, 75.1)[a,b] | **0.000** |
| Recording time with SpO2 <90% | / | 0.00(0.00, 0.50)[c] | 0.70(0.10, 6.80)[c] | 62.75(23.50, 146.43)[a,b] | **0.000** |
| Lowest nocturnal SpO2% | / | 92.00(87.00, 92.00)[c] | 86.00(84.00, 90.00)[c] | 74.50(61.75, 79.25)[a,b] | **0.000** |
| Mean nocturnal SpO2% | / | 95.00(95.00, 96.00)[c] | 95.00(94.00, 95.00)[c] | 93.00(91.00, 94.00)[a,b] | **0.000** |
| **PSG monitoring** | | | | | |
| Total Sleep Time minutes | / | 450.40(355.90, 476.50) | 456.50(431.40, 484.10) | 454.35(396.85, 477.65) | 0.623 |
| Sleep Latency minutes | / | 8.50(4.00, 29.50) | 7.00(3.50, 15.50) | 7.00(3.38, 11.50) | 0.713 |
| REM Period of Sleeping minutes | / | 81.50(51.00, 103.00) | 86.50(66.50, 103.50) | 73.00(56.88, 94.75) | 0.244 |
| NREM Period of Sleeping minutes | / | 354.00(303.40, 396.40) | 380.00(325.00, 394.30) | 368.40(331.45, 402.73) | 0.475 |
| SWS Period of Sleeping minutes | / | 0.00(0.00, 8.00) | 0.00(0.00, 19.00) | 0.00(0.00, 4.75) | 0.678 |
| MicroArousal Index | / | 11.50(7.20, 18.30) | 13.10(10.10, 19.30) | 36.00(24.33, 49.68) | **0.000** |
| **Blood pressure mmHg** | | | | | |
| Systolic | 114.0(105.5, 124.0)[d] | 110.0(100.0, 125.0)[d] | 110.0(105.0, 116.0)[d] | 121.5(111.0, 135.25)[a,b,c] | **0.002** |
| Diastolic | 70.0(65.0, 76.0)[d] | 72.0(63.0, 87.0) | 72.0(65.0, 79.0)[d] | 80.5(74.5, 87.0)[a,c] | **0.000** |
| **White cell count cells mm $-3$** | | | | | |
| Total | 5.45(4.75, 6.16)[d] | 5.49(4.63, 6.96) | 5.94(5.37, 6.60) | 6.09(5.33, 7.18)[a] | **0.004** |
| Neutrophils | 2.99(2.54, 3.63)[d] | 3.14(2.41, 3.79) | 3.55(2.69, 3.93) | 3.34(2.89, 4.23)[a] | **0.033** |
| Lymphocytes | 1.78(1.50, 2.13)[a] | 1.88(1.51, 2.33) | 1.84(1.57, 2.33) | 2.10(1.71, 2.37)[d] | **0.041** |
| Monocytes | 0.38(0.33, 0.48) | 0.36(0.30, 0.51) | 0.38(0.32, 0.48) | 0.41(0.32, 0.48) | 0.651 |
| Neutrophil to Lymphocyte Ratio | 1.76(1.29, 2.14) | 1.70(1.25, 2.18) | 1.73(1.36, 2.43) | 1.68(1.37, 2.41) | 0.910 |
| **Haemoglobin g dL $-1$** | 138.47 ± 15.08[d] | 141.43 ± 10.60[d] | 143.87 ± 16.83 | 153.30 ± 16.76[a,b] | **0.000** |
| **Platelet count** | 214.00(181.50, 249.50) | 233.00(189.00, 253.00) | 223.00(183.00, 245.00) | 239.50(181.25, 275.50) | 0.369 |
| Platelet to Lymphocyte Ratio | 119.12(95.96, 151.09) | 114.35(92.96, 149.68) | 109.40(95.71, 143.78) | 111.13(85.43, 139.97) | 0.613 |
| **T lymphocyte count** | | | | | |
| CD3+T | / | 1192.00(923.00, 1588.00) | 1208.00(961.00, 1587.00) | 1264.00(1060.75, 1646.75) | 0.474 |
| CD4+T | / | 628.00(477.00, 696.00) | 620.00(5116.00, 746.00) | 642.50(505.50, 771.50) | 0.748 |
| CD8+T | / | 443.00(35.200, 551.00) | 449.00(332.00, 605.00) | 455.00(369.00, 596.25) | 0.886 |
| CD4+T to CD8+T Ratio | / | 1.31(0.91, 2.06) | 1.27(1.01, 1.64) | 1.36(0.88, 1.81) | 0.903 |
| **Total Antibody** | | | | | |
| IgA | / | 2210.00(1880.00, 2430.00) | 2540.00(2120.00, 3090.00) | 2425.00(1795.00, 3025.00) | 0.154 |
| IgG | / | 11.50(10.30, 12.80) | 12.10(10.80, 13.80) | 12.00(10.70, 13.95) | 0.620 |
| IgM | / | 1030.00(784.00, 1460.00) | 1300.00(922.00, 1630.00) | 1090.00(838.00, 1790.00) | 0.377 |

**Notes.**

Significant different from: HD (a), Non-OSA (b), Mild-to-mderate OSA (c), Severe OSA (d).

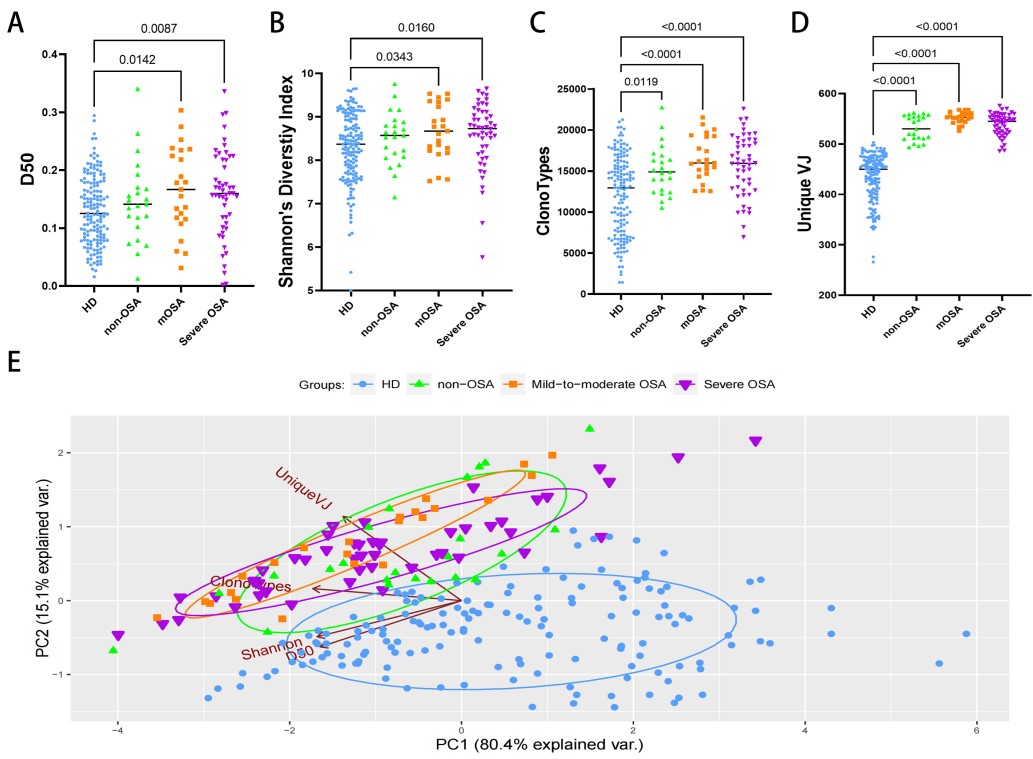

**Figure 1   The diversity and clonalities analysis of expressed TCRβ repertoire.** Plots diagram showing the D50 values (A), the Shannon's Diversity Index (B), total clonotypes (C), and unique VJ counts (D) of repertoires in each group. Principal component analysis (E) depicts each sample on the abundance of TCR clones. The distance between dots indicates the degree of dissimilarity of the TCR repertoire profile between samples. mOSA, mild-to-moderate OSA.

compared to the healthy control group (Figs. 1A, 1B; Table S1), suggesting a putative increase in clonotypes in OSA patients. Furthermore, comparison of clonotypes and unique VJ counts between the healthy control and other groups confirmed that patients with OSA and other non-OSA sleep disturbances are impaired in TCR diversity and clonality (Figs. 1C, 1D; Table S1). We further performed the principal component analysis and observed that the sample plots from the OSA patients were remarkably clustered to the upper left region in the picture, significantly different from those from healthy donors (Fig. 1E).

## Characteristics of V, J gene usage, and amino acid clonotypes

We then evaluated the V and J gene usage of TCRβ CDR3 repertoire in severe OSA compared to other groups (Figs. 2A, 2B, 2C; Fig. S3) and identified a series of V, J gene usage and V-J alleles that were significantly different between severe OSA and other groups (Table S2). Mild-to-moderate OSA also showed a significant difference in V, J usage and combination from Non-OSA (Fig. 2D; Fig. S4). As shown in volcano plots (Fig. 3), the expression of amino acid clonotypes differed between every two groups (Table S3). Remarkably, the clonotype of ASSSGSSYNEQF was significantly decreased in severe OSA.

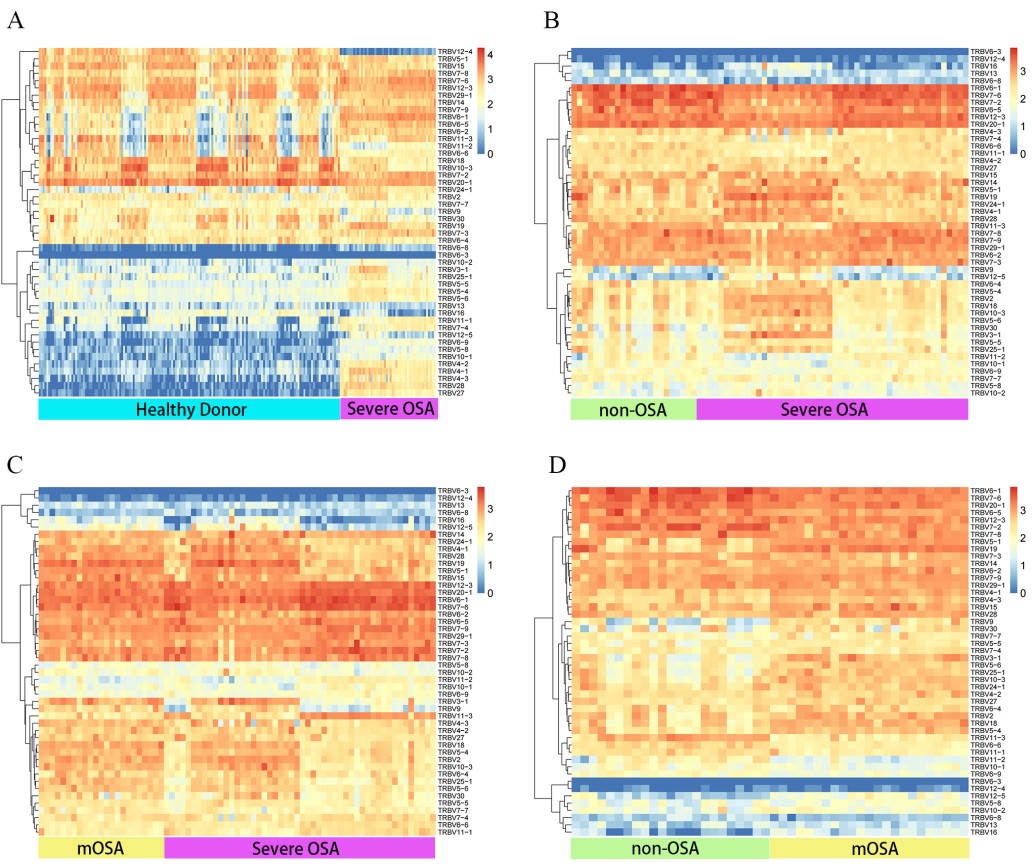

**Figure 2** **Distinct Vβ gene usage in comparison to all groups.** Heatmap showing the frequencies of Vβ expression in comparison between the following groups: severe OSA and healthy donors (A), severe OSA and non-OSA (B), severe OSA and mild-to-moderate OSA (C), mild-to-moderate OSA and non-OSA (D). The color represents the log10 value of gene expression in each block. mOSA, mild-to-moderate OSA.

Moreover, the clonotype of ASGPREK showed a lower expression in both severe OSA groups and mild-to-moderate OSA.

## Validation of developed model

Accordingly, we termed a novel indicator OSA-TCI by computer screening to evaluate whether it could help distinguish severe OSA patients from others. In training and leave-one-out cross-validating, the optimal algorithm achieved an AUC of 0.974 (95% CI [0.930–1.000]; Figs. 4A, 4B). In separate testing validation, its AUC yielded 0.872 (95% CI [0.745–0.999]; Figs. 4C, 4D). In order to compare with previously reported biomarkers in other studies (*Rha et al., 2020*; *Domagała-Kulawik et al., 2015*; *Lévy et al., 2015*; *Wu et al., 2018*), we drew a ROC for each variable using the data of all 96 patients who finished PSG, in which 63 samples were not ever used in model training. The AUC of OSA-TCI was 0.914 (95% CI [0.853–0.975]; Fig. 4E, Table S4), much better than BMI, ESS score, NLR, PLR, and CD4+/CD8T ratio. With a cutoff value of 1087, OSA-TCI achieved a sensitivity of 90% and a specificity of 87% in discriminating severe OSA from participants of PSG. The

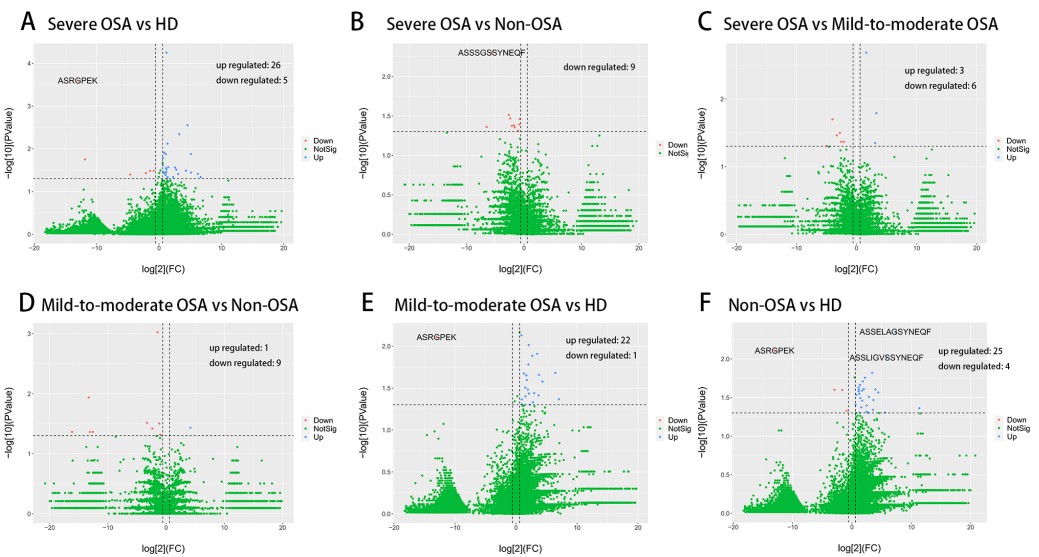

**Figure 3** **Comparison of amino acid clonotypes.** Volcano plots showing the different expression (log2(FC) > 1, $p < 0.05$) of amino acid clonotypes between severe OSA and healthy donors (A), severe OSA and non-OSA (B), severe OSA and mild-to-moderate OSA (C), mild-to-moderate OSA and non-OSA (D), mild-to-moderate OSA and healthy donors (E), non-OSA and healthy donors (F).

distinction of OSA-TCI between severe OSA and other groups promised the possibility of being a potential biomarker for severe OSA.

## Rationalize OSA-TCI with clinical features

AHI, BMI, hemoglobin concentration, the percentage of N1, N2 stage sleeping duration, snoring, smoking, and maximum nocturnal oxygen desaturation was revealed to be significantly associated with OSA-TCI by Spearman correlation analysis for two variables (Table S5). While, by stepwise backward strategy on linear regression, we identified only AHI ($P < 0.001$) and smoking ($P = 0.032$) rather than other variables were independent associate factors of OSA-TCI. The Spearman correlation coefficient between OSA-TCI and AHI is $R = 0.603$ (P =0<0.001), and the relation to smoking is $R = 0.319$ ($P = 0.002$). Finally, we examined all variables using partial coefficient analysis controlled by AHI (Table S6) and found that only smoking remain independently related to OSA-TCI, $R = 0.22$ ($P = 0.032$). Thus, the data analysis demonstrated that AHI and smoking were independent contributors to the OSA-TCI value.

## TCR repertoire reflected therapeutic effects of nPAP in follow-up consistently

NPAP therapy is generally acknowledged as the first choice for severe OSA patients. In the follow-up, eight severe OSA patients had completed 3 months' family non-invasive ventilation therapy of good adherence (>4 hrs per night, >20 hrs per week). We compared their TCR repertoires and observational indexes before and after treatment (Fig. S5, Table S7): Their TRBV and TRBJ usage significantly changed after nPAP (Figs. 5A, 5B). Notably, regardless of ESS level before treatment, the OSA-TCIs of all eight patients were consistently

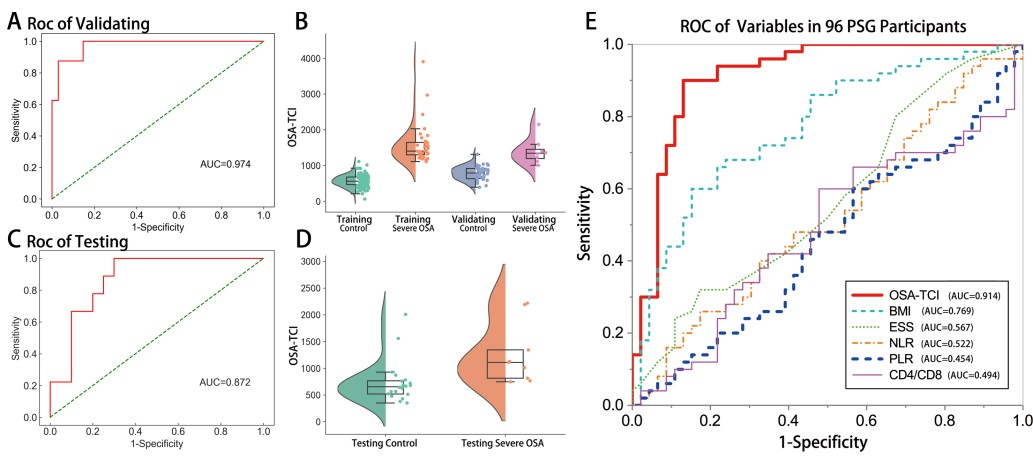

**Figure 4 Validation of OSA-TCI as an indicator of severe OSA.** Applying our optimal machine-developed algorithm, the ROC (A) and semi-violin plots (B) of validating group ($n = 42$), the ROC (C) and semi-violin plots (D) of the testing group ($n = 29$), and the comparison of ROC performances of OSA-TCI, BMI, ESS, NLR, PLR, CD4+/CD8+T ratio (E) in all PSG participants ($n = 96$) exhibit the potential diagnostic power of OSA-TCI.

reduced and statistically different ($P = 0.0078$), and the values decreased greater in patients with higher OSA-TCI (Fig. 5C). Meanwhile, the D50 values and Shannon's diversity index of treated individuals showed different degrees of increase with statistical difference (Table S7, Fig. S6). At the same time, the overall ESS also decreased after ventilator treatment, while some patients with low ESS scores did not improve further, and no significant difference was found with regard to the small sample size (Fig. 5). Additionally, we did not observe any statistical difference in NLR and CD4+/CD8+T before and after the treatment. The inconsistent trend shown by the figure could not directly reflect the effect of ventilator therapy on immune cell counts (Figs. 5E, 5F).

## DISCUSSION

Previously, numerous studies have demonstrated the relationship between OSA severity and worse clinical consequence in comorbidities (*George, 2007*; *McNicholas et al., 2018*; *Bradley & Floras, 2009*; *Justeau et al., 2020*; *Rögnvaldsson et al., 2022*; *Lavie, 2015*). Meanwhile, an increase in circulating pro-inflammatory cytokines and oxidative products such as IL-6, TNF-a, CRP, NFkB, and cell-free DNA peptides were also observed to correlate positively with AHI (*Yokoe et al., 2003*; *Ryan, Taylor & McNicholas, 2005*; *Cubillos-Zapata et al., 2017*; *Bauça et al., 2017*). Moreover, effective therapy for severe patients, such as the implementation of adherent nPAP or surgery, can not only lower the inflammatory biomarker levels (*Yokoe et al., 2003*; *Xie et al., 2013*) but also significantly improve endothelial function and microvascular blood flow (*Xu et al., 2015*), resistant hypertension (*Iftikhar et al., 2014*), sleepiness, quality of life, mood, neurocognitive function, left ventricle ejection fraction, hemoglobin A1c, fasting glucose, cardiovascular events incidence, and incident mortality (*Patil et al., 2019*). The inflammatory nature is therefore supposed to be one of the essential mechanisms leading to systemic injury and

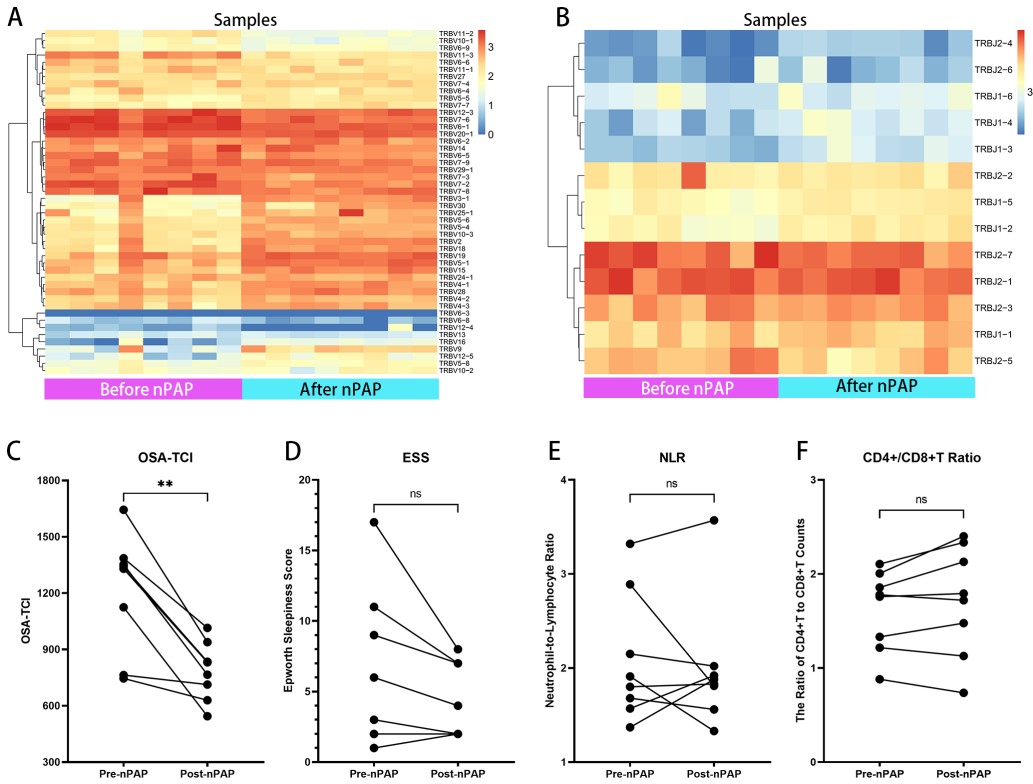

**Figure 5 Therapeutic effects of nPAP reflected by TCR repertoire and biomarkers.** In follow-up of eight nPAP treated patients with severe OSA: their Vβ (A) and Jβ (B) gene expressions after therapy are distinct from before. Their OSA-TCI value (C) consistently reflects the significant change, but with no substantial differences in ESS (D), NLR (E), or CD4+/CD8+T ratio (F) after therapy. The asterisks indicate *p*-values of the paired sample Wilcoxon signed ranks test (**: $p < 0.01$).

comorbidity deterioration in OSA (*Rögnvaldsson et al., 2022*; *Lavie, 2015*; *Ryan, Taylor & McNicholas, 2005*), valuable in disease surveillance and treatment. Therefore, interest focus on exploring biomarkers has been increasing for decades. In recent years, additional research on cellular immune has reported that macrophage polarization towards M1 subsets (*Khalyfa, Kheirandish-Gozal & Gozal, 2018*), increased ratio of NLR (*Rha et al., 2020*), PLR (*Kıvanc et al., 2018*), CD8+T/CD4+T (*Domagała-Kulawik et al., 2015*), and Th17/Treg (*Reale et al., 2020*), as well as DNA methylation of toll-like receptor 2/6 (*Huang et al., 2020*) in OSA patients, suggesting possible reactions to new endogenous antigens. Given the vital role of T cells in adaptive immunity, the question arises as to why and how T cell immunity is impaired in OSA.

This study provides the first investigation into TCR repertoire features in patients with OSA towards the question above. Remarkably, by sequencing, we revealed that notable disease-severity-associated clones, disease-preferred V *β*, J *β*, and V-J junctions exist in OSA and diversity and new clonotypes dramatically increased, suggesting unusual T cell activation and memory for a series of unique MHC-peptides complex exposure. By contrast, a significant decrease in diversity is a common phenomenon that can be found

in most previous studies on other conditions, such as cancer, tuberculosis (*Fu et al., 2019*), pneumonia of COVID-19 (*Chang et al., 2021*), acute myocardial infarction (*Zhong et al., 2019*), and subarachnoid hemorrhage (*Kim et al., 2020*) and corresponds with activation of specific associate TCR clusters to emergent stimuli. Notably, there are several significant factors that can affect the TCR repertoire diversity: (1) aging: the thymus begins to wither after puberty, gradually losing naïve T lymphocytes production, decreasing the ability to respond to neoantigens with age (*Douek et al., 1998*). Our data also shows significant D50 declination with age in OSA patients and healthy donors; (2) immune response against antigens: clonal expansion after T cells activation can remarkably change the TCR repertoire diversity (*Chen et al., 2017*). After the elimination of antigens, the expanded TCR clones will decline, and only a tiny portion of these expanded TCR clones will be kept in the memory T cells department. (3) Death and exhaustion of T cell subsets. (4) Immune regulation: the crosstalk of immune cells, cytokines, and interferon molecular can modulate certain groups of T cell subsets. Importantly, HIF-1 is a ubiquitously expressed critical modulator in T cell survival, proliferation, and differentiation by altering the metabolic strategy of immune cells (*Taylor & Colgan, 2017*; *McGettrick & O'Neill, 2020*). Thus, impaired oxygen availability by CIH, which intensively promotes HIF-1$\alpha$ production, may bring a wide range of pro-inflammatory and anti-inflammatory implications to a series of immune cells. OSA patients, featured by the unique pattern of replicated nocturnal CIH, may induce more extensive, long-lasting, and multi-elements drove adaptive alteration rather than uniformed clonal expansion induced by acute massive stimuli. Furthermore, long-term recurrent exposure to inflammatory conditions may also induce immune tolerance to maintain homeostasis, thus reducing the probability of large clones.

Of note, the adaptive immunity impairment, pre-existing inflammatory status, and frequent fluctuation of oxygen saturation overnight also mean additional danger for OSA patients during the pandemic. The reported OR for death from severe COVID-19 in OSA patients varies from 2.0 to 2.8 compared with non-OSA, when adjusted for demographic characteristics and commodities (*Rögnvaldsson et al., 2022*; *Cariou et al., 2020*), with no data for severe OSA patients of higher risk. Besides, many risk factors for OSA are shared in severe COVID-19 patients, several possible mechanisms may lead to poorer outcomes for OSA: first, the preset chronic inflammation status may predispose to an imbalanced pro-inflammatory response to infections, more susceptible to immune factor storms (*Ryan, Taylor & McNicholas, 2005*; *Taylor & Colgan, 2017*); second, the hypoxemia in severe COVID-19 patients can be further exaggerated by CIH level associated with OSA severity, leading to more severe hypoxia-reoxygenation stress injury and cascade reactions. Third, recurrent CIH leads to a high level of HIF (*Taylor & Colgan, 2017*; *McGettrick & O'Neill, 2020*), which plays a vital role in immune dysfunction, results in macrophage polarization, neutrophil apoptosis, T cell subsets alteration, as well as the TCR repertoire changes, and may impair the ability to eliminate pathogens and sustain immunity balance. Unfortunately, on the other hand, the COVID-19 pandemic has caused a worldwide dramatic shrink in accessibility for in-lab diagnosis for fear of contamination. An investigation (*Grote et al., 2020*) reported that compared to pre-pandemic levels, there was a significant reduction in PSG by 20% on average, with staffing levels reduced to 25% of sleep physicians and

19% for nurses and technicians, in nineteen European countries. In order to screen out substantial severe OSA patients for necessary treatment, an easy-access and accurate tool with the lowest contaminate risk is called for in urgent need.

Through machine learning of disease-specific clones, we developed a novel indicator termed OSA-TCI to metric the extent of adaptive immune dysfunction related to severe OSA. Afterward, separate sample validation identified the overall diagnostic power of OSA-TCI to a sensitivity of 90% and a specificity of 87% in distinguishing severe OSA patients from others, with an optimal cut-off of 1087, AUC = 0.914. Although it has been reported that NLR, PLR, and CD4+/CD8+T ratio can serve as a biomarker for OSA in diagnosis (*Rha et al., 2020*; *Domagała-Kulawik et al., 2015*; *Kıvanc et al., 2018*), these studies neither provided any necessary validation in separate samples nor exhibited the AUC of ROC; the ROC in our study failed to support their diagnostic value for the sake of their poor AUCs. Thus, there was no reliable method for diagnosing and evaluating OSA from peripheral blood before our study. As a novel approach to identifying biomarkers, TCR immune repertoires were previously proved efficient in identifying infectious diseases, autoimmune diseases, carcinoma, and even acute myocardial infarction (*Fu et al., 2019*; *Chang et al., 2021*; *Zhong et al., 2019*; *Kim et al., 2020*). The complex but unique chronic inflammatory features of OSA, including CIH-induced cell injury and apoptosis products, endothelial function impairment, abnormal metabolomics, colonized microbial flora changes, the impacts of sleep fragmentation, and even early lesion in the cardiovascular system (*Lavie, 2015*; *Khalyfa, Kheirandish-Gozal & Gozal, 2018*; *Humer, Pieh & Brandmayr, 2020*), can probably contribute to forming a disease-specific repertoire landscape. There are also intimate and complex interactions between CIH-induced metabolic disturbances, immune pathway shifts that affect immune cell function, and TCR repertoire successfully integrated the picture to some extent. Although immune sequencing is not economical enough currently, some scientists are exploring cheaper TCR CDR3 sequencing methods, such as Framework Region 3 Amplifi-Kation sequencing (*Amoriello & Ballerini, 2020*), for better utilization in a large population. Furthermore, another advantage of TCR repertoire-based assessment is that by applying previously established classifiers for a series of diseases, many conditions can be identified simultaneously from just a single blood sample (*Arnaout et al., 2021*).

According to the rationalization of OSA-TCI with clinical features, OSA-TCI correlates positively with AHI, $R = 0.603$, demonstrating that CIH may be the dominant contributor to TCR repertoire alteration. Smoking is considered to be another significant contributor, and $R = 0.22$ (adjusted for AHI) hints at the putative linkage between OSA-TCI and endothelial injury aside from AHI, as smoking is also an evidenced independent risk factor of OSA, endothelial function impairment, hypertension, coronary disease. In correlation analysis of PSG participants, BMI is clarified as a confounding factor that contributes a weak correlation to OSA-TCI, $R = 0.342$, $P = 0.001$, but when BMI is adjusted for AHI, $R = 0.03$, $P = 0.772$. In contrast, no significant relationship between OSA-TCI with TST, SWS sleep, REM sleep, excessive daytime sleepiness, NLR, PLR, CD4+/CD8+T ratio, or lymphocyte counts. Whereas in individuals, OSA-TCI is not exactly in a linear relationship with AHI, which may reflect the differences in hypoxia tolerance and immuno-metabolism between

individuals, which can not be explained by AHI alone, but may bring about variable impacts on clinical prognosis and treatment effect. AHI, as a single biomarker calculated from one night of PSG, can reflect the frequency of respiratory events during sleep but is unable to accurately grasp the abnormalities underlying its immune, metabolic, and cardiovascular effects on each individual (*Malhotra et al., 2021*), and sometimes were biased by night-to-night variation or incorrect scoring. Compared to AHI, OSA-TCI is acquired from the TCR repertoire characteristic algorithm, which involves multiple individualized responses to CIH and immuno-metabolic alterations, and may have the ability to predict prognosis and therapeutic effects. To our delight, the follow-up data indicate that, in all eight severe OSA patients who accepted adherent nPAP treatment for three months (>4 hrs per day), OSA-TCI remarkably decreased compared with their baselines. With the coincident reduction of OSA-TCI, their improvement in TCR repertoire and amelioration of daytime sleepiness and fatigue suggests its potential capability for treatment assessment. Of interest, from our small sample of before-and-after data, it appears that patients with severe OSA who had higher OSA-TCI improved more with standardized nPAP therapy. Additionally, we observed the D50 value and Shannon's Diversity index increased after treatment, for which the mechanism is not clear and may be related to the alleviation of CIH induced immune disorders, although these two indicators are not disease-specific. Before this study, only ESS was reported valuable in predicting therapeutic effects of nPAP (*Malhotra et al., 2021*; *Marshall et al., 2006*), but it seems OSA-TCI is a more sensitive biomarker, especially in some severe patients with low sleepiness scores. With an ongoing follow-up of severe OSA in the cohort, we will test our hypothesis in a larger sample amount whether patients with higher OSA-TCI may suffer a higher risk of complications and worse clinical consequences but benefit more from therapies.

Limitations of this study include no paired TCR alpha chain sequencing, no flow cytometry to sort T cell subsets, limited blood indicators, short follow-up duration, and lack of PSG on healthy controls. In light of these limitations, subsequent research should involve single-cell sequencing of circulating lymphocytes from OSA to obtain more insights. A blinded prognostic cohort is also necessary in confirming the predictive value of OSA-TCI. Currently, economic expenditure may hinder the utilization of TCR repertoire in clinical practice, therefore, we conducted direct sequencing of TCR without previous flow cytometry to make the strategy cheaper and easier to use. However, some inflammation indexes, such as CRP, IL-6, IL-10, and TNF-$\alpha$, were not considered a potential biomarker for OSA in our study for their poor specificity.

Above all, our study unveiled that the TCR repertoire changes remarkably in OSA patients, especially in severe patients, suggesting that an underlying auto-immune mechanism is closely related to intermittent hypoxia, smoking and proposed a convenient indicator for disease identification and surveillance without long-time contact and facility/instrument occupation. Furthermore, as an essential aspect of immune dysfunction in OSA, the TCR repertoire may be helpful in future therapeutic monitoring, complication risk prediction, and immunological and pathophysiological study.

## ACKNOWLEDGEMENTS

We owe our thanks to our colleagues in the Department of Respiratory and Critical Care Medicine, the Sleep Medicine Centre, and the Health Management & the Institute of Health Management who provided expertise that greatly assisted the research.

### Funding

This research was supported by the Sleep Medicine Center, West China Hospital, Sichuan University and Sichuan Provincial People's Hospital, University of Electronic Science and Technology of China. The funders had no role in study design, data collection and analysis, decision to publish, or preparation of the manuscript.

### Grant Disclosures

The following grant information was disclosed by the authors:
Sleep Medicine Center, West China Hospital, Sichuan University.
Sichuan Provincial People's Hospital, University of Electronic Science and Technology of China.

### Competing Interests

Zhixin Zhang is the founder of Chengdu ExAb Biotechnology, LTD. Xin Yang and Xueping Wen are employed by Chengdu ExAb Biotechnology LTD.

### Author Contributions

- Kai Li conceived and designed the experiments, performed the experiments, analyzed the data, prepared figures and/or tables, authored or reviewed drafts of the article, and approved the final draft.
- Yue Zhuo performed the experiments, analyzed the data, prepared figures and/or tables, authored or reviewed drafts of the article, and approved the final draft.
- Yue He analyzed the data, authored or reviewed drafts of the article, and approved the final draft.
- Fei Lei performed the experiments, prepared figures and/or tables, and approved the final draft.
- Pengming He analyzed the data, prepared figures and/or tables, and approved the final draft.
- Qin Lang performed the experiments, prepared figures and/or tables, and approved the final draft.
- Dingxiu He performed the experiments, prepared figures and/or tables, and approved the final draft.
- Suni Zuo performed the experiments, authored or reviewed drafts of the article, participants recruitment, clinical data management, and approved the final draft.
- Shan Chen performed the experiments, authored or reviewed drafts of the article, participants recruitment, clinical data management, and approved the final draft.

- Xin Yang performed the experiments, analyzed the data, authored or reviewed drafts of the article, and approved the final draft.
- Xueping Wen performed the experiments, analyzed the data, authored or reviewed drafts of the article, and approved the final draft.
- Zhixin Zhang analyzed the data, authored or reviewed drafts of the article, and approved the final draft.
- Chuntao Liu conceived and designed the experiments, authored or reviewed drafts of the article, and approved the final draft.

## Human Ethics

The following information was supplied relating to ethical approvals (i.e., approving body and any reference numbers):

The experimental design and participant recruitments were approved by the Ethics Committee of West China Hospital, Sichuan University (Ethics Approval for Research: 2021 #1302) and the Medical Ethics Committee of Sichuan Provincial People's Hospital (Ethics Approval for Research: 2020 #351).

## Ethics

The following information was supplied relating to ethical approvals (i.e., approving body and any reference numbers):

The experimental design and participant recruitments were approved by the Ethics Committee of West China Hospital, Sichuan University (Ethics Approval for Research: 2021 #1302) and the Medical Ethics Committee of Sichuan Provincial People's Hospital (Ethics Approval for Research: 2020 #351).

## Data Availability

The processed functional TCR$\beta$ sequence data analyzed in this study are available under National Genomics Data Center (NGDC) Genome Sequence Archive (GSA): HRA003311.

## Supplemental Information

Supplemental information for this article can be found online at http://dx.doi.org/10.7717/peerj.15009#supplemental-information.

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
