# Peer review of "T cell receptor repertoire as a novel indicator for identification and immune surveillance of patients with severe obstructive sleep apnea"

_PeerJ, doi:10.7717/peerj.15009_

## Round 0.1 · original submission · Minor Revisions

The reviewers have provided several important comments, which could improve the quality of manuscript.

Reviewer 1 ·

Basic reporting

Despite some small typos and mistakes, the manuscript was well-written using professional and easy-understandable language. The introduction offered a clear overview of the current knowledge of obstructive sleep apnea and the aim of the study. However, in the Material and methods part, the details of how the TCR sequencing library is prepared are missing (line 143 to 148). It was unclear whether there is a T cell enrichment step or a CD4+/CD8+ isolation step. It is ambiguous whether the TCR repertoire was from CD4+ T cells or CD8+ T cells or both.
In general, the structure of the manuscript was well-arranged. By discussing the differences in TCR repertoires between different OSA patient groups and healthy donors, the authors were able to find a correlation between OSA status and TCR repertoire diversity. The figures are relevant, high quality, and well-labeled and described. The raw data was supplied.

Experimental design

The experimental design was straightforward and easy to follow. However, the result was mostly about correlations between TCR repertoire and OSA status. The authors did not further investigate the mechanism behind the correlation which makes the conclusion a little weak. In addition, in the results part, the discussion about their findings on the clonotypes can be further elaborated (line 252 to 254). For example, whether there are similar findings in other studies or diseases.

Validity of the findings

The finding of the manuscript is novel and can serve as a reference for further research on OSA and T cells. The conclusion and the limitations of the research were both clearly stated and linked to the original research questions.

Additional comments

The lack of cell sorting prior to sequencing and mechanism proof beyond correlation weakens the conclusion. Moreover, the authors didn't mention the status of the MHC genotype of the donors which can be a confounding factor for the differences in TCR repertoire. But nevertheless, the authors still made a fair conclusion from the data they generated and successfully avoided over-explaining.

Annotated reviews are not available for download in order to protect the identity of reviewers who chose to remain anonymous.

Reviewer 2 ·

Basic reporting

1. The overall writing is clear.
2. Line89: 'speci city', please correct the word.
3. Line91: 're ect', please correct the word.
4. Tabe1, page27: 'T cell lymphocyte cout', please correct the word.

Experimental design

Since the modeling dataset is unbalanced, which has more healthy samples compared to OSA patients. The author should try some methods to balance the modeling dataset and validating the performance of their model on validation and testing dataset.

Validity of the findings

Is there any improvement of the D50 value and Shannon's Diversity index after nPAP treatment?

Additional comments

no comments

---

## Round 0.2 · accepted · Accept

Thank you very much for your careful response. The reviewer's comments have been well addressed.